# Biological Control of *Pseudomonas syringae* in Tomato Using Filtrates and Extracts Produced by *Alternaria leptinellae*

**Carlos García-Latorre** [1] , **Sara Rodrigo** [2] **and Oscar Santamaria** [3,*]

1   School of Agricultural Engineering, University of Extremadura, Avda. Adolfo Suárez s/n,
    06007 Badajoz, Spain; cgarcialn@unex.es
2   Institute of Dehesa Research (INDEHESA), University of Extremadura, Avda. Elvas s/n, Campus de Badajoz,
    06006 Badajoz, Spain; saramoro@unex.es
3   Department of Plant Production and Forest Resources, Sustainable Forest Management Research
    Institute (iuFOR), University of Valladolid, Avda. Madrid 57, 34004 Palencia, Spain
*   Correspondence: oscar.santamaria@uva.es

**Abstract:** Endophytic fungi offer promising alternatives for sustainable plant disease management strategies, often through the production of bioactive secondary metabolites. This study investigated the biocontrol potential of filtrates and extracts, produced under controlled conditions, from *Alternaria leptinellae* E138 against *Pseudomonas syringae* in tomato plants under greenhouse conditions. To understand the main mechanisms involved in biocontrol, the direct inhibition of bacterial growth and disruption of quorum sensing activity caused by metabolites were studied in vitro, as well as indirect mechanisms, such as their capacity to produce phytohormone-like substances, nutrient mobilization, and antioxidant activity, which can enhance plant growth and fitness. Moreover, a mass spectrometry analysis was used to tentatively identify the secondary metabolites present in the extract with antimicrobial properties, which could explain the biocontrol effects observed. Mycopriming assays, involving the direct treatment of tomato seeds with the fungal *A. leptinellae* E138 extracts, produced increased germination rates and seedling vigor in tomato seeds. As another treatment, postemergence application of the extracts in greenhouse conditions significantly improved plant health and resulted in a 41% decrease in disease severity. Overall, this study underscores the potential of *A. leptinellae* E138 extract as a plant growth promoter with biocontrol capabilities, offering promising avenues for sustainable plant disease management.

**Keywords:** fungal endophytes; biological control; plant growth promotion; metabolites; sustainable agriculture; *Alternaria*; *Pseudomonas*



## 1. Introduction

Tomato (*Solanum lycopersicum* L.) is a significant agricultural crop globally, serving as a vital source of food, income, and exports across many regions, including Europe, East Africa, and Indonesia [1,2]. Its economic importance is underscored by global production of 189.1 million t and cultivation on 5.1 million ha in 2021 [3], due to its widespread consumption and utilization in the food industry [4]. However, the tomato crop faces numerous challenges, including diseases, pest infestations, and environmental stressors. Consequently, comprehensive research and management strategies are required to ensure sustainable tomato production, thereby safeguarding food security and economic stability [5,6]. Among the challenges faced, tomato is notably susceptible to the pathogen *Pseudomonas syringae* pathovar tomato, the causal agent of the bacterial speck (BS) disease, which has been associated with substantial crop losses in regions like Europe and Mediterranean Africa [7]. BS disease manifests as small, dark lesions with a speck-like appearance on tomato leaves, stems, and fruit [8], often merging to cause extensive tissue damage [9]. In severe cases, infected fruit can become unmarketable due to their low quality and unsightly appearance [10]. Thus, the economic impact caused by BS of tomato can be

profound, resulting in substantial yield losses and rendering the affected fruit unsuitable for both fresh consumption and processing.

Conventional disease control methods primarily involve synthetic chemicals, posing environmental hazards when applied indiscriminately. More novel and environmentally friendly strategies to reduce the impact of this pathogen include the implementation of varietal resistance or epidemiological modeling. Notably, the utilization of fungal endophytes as biocontrol agents (BCAs) to protect tomato plants against pathogens, such as *P. syringae*, has garnered attention due to their potential to enhance plant defense mechanisms and promote growth. Fungal endophytes, residing within plant tissues without causing apparent harm to the host [11], play a crucial role in enhancing plant tolerance to biotic and abiotic stresses, including pathogen defense [12–14]. For instance, Amruthesh et al. [15] demonstrated that fungal endophytic seed treatment induced disease resistance against early blight in tomatoes. Similarly, Constantin et al. [16] found that endophyte-mediated resistance (EMR) against Fusarium wilt disease in tomatoes was independent of certain defense signaling pathways, indicating the potential of endophytic fungi in enhancing plant resistance.

However, the effective use of fungal endophytes as BCAs requires a comprehensive understanding of the endophyte–host relationship, endophyte–pathogen interactions, and the influence of environmental factors on these dynamics [17,18]. The nature of the endophyte-host relationship may vary across tomato cultivars, highlighting the importance of considering host specificity in endophyte selection for biocontrol purposes [19]. Furthermore, the adaptability of endophytes to the edaphoclimatic conditions of the application site compared to those of their isolation is crucial [20]. These factors may affect the efficacy and suitability of direct application of these endophytes for disease control. The utilization of the secondary metabolites produced by the endophyte, rather than the living organism, could provide greater efficacy and mitigate such challenges, as these metabolites have often been identified as the primary agents responsible for biocontrol activity [21,22], and may not be so influenced by host specificity or edaphoclimatic conditions. The application of these bioactive compounds produced in vitro has shown to be promising not only in biocontrol activity but also in promoting various plant growth traits such as germination, vigor, and chlorophyll content, highlighting their potential to enhance overall plant health [23]. Additionally, fungal filtrates have been found to be effective in controlling certain tomato crop pathogens, such as *Alternaria solani* [24]. These compounds also offer the advantage of generally being non-toxic and posing minimal environmental risk, though further evaluation is necessary [25]. Furthermore, they can be easily and inexpensively produced, making them a cost-effective solution for tomato farmers.

In the specific case of *P. syringae*, its biocontrol using fungal endophytes has been demonstrated in various plant species [26,27], including the use of their fungal filtrates [28], which promote plant growth and stimulate plant resistance. Hence, the hypotheses of this study were (1) that the biocontrol activity of a fungal endophyte species (if any) is mainly attributed to the secondary metabolites it produces, and (2) that the filtrates/extracts produced in vitro by this endophyte, which may contain such metabolites, could protect plants against pathogens when applied externally. To test these hypotheses, the main goal of the present study was to assess whether filtrates or extracts of *Alternaria leptinellae*, a fungal endophyte previously isolated from healthy plants and demonstrated to possess antimicrobial activity, could effectively control *P. syringae* upon application to tomato seeds and seedlings. Furthermore, to gain insight into the mechanisms underlying potential protection, several traits of the endophyte and its filtrates/extracts that could directly or indirectly affect plant protection—such as direct antagonism with bacteria, phytohormone production, antioxidant activity, or nutrient solubilization activity—were also evaluated in vitro.

## 2. Materials and Methods

### 2.1. Fungal and Plant Material

The fungal endophyte used in this study was previously isolated from healthy leaves of *Ornithopus compressus* plants growing in the *dehesas* of Extremadura, southwestern Spain. The fungus, designated with the internal code E138, was identified as *Embellisia leptinellae* E.G. *Simmons & C.F. Hill* (Thom), the basionym of *Alternaria leptinellae*. Initial identification involved a morphological assessment of its reproductive structures, followed by a molecular characterization through comparison of its ITS region sequence with entries in two databases, GenBank (www.NCBI.nlm.nih.gov) and UNITE (https://unite.ut.ee), employing a BLAST search method [29]. Consequently, the isolate was assigned the Genbank accession number KP698337 [30]. Selection criteria for this species included its high frequency of isolation from the original plant host and promising bioactivity observed in previous assays [31,32].

In vitro antagonism tests were conducted using two bacterial strains, irrespective of their pathogenicity: Gram-positive *Bacillus subtilis* NCTC 8236 and Gram-negative *Pseudomonas syringae* pv. tomato NCPPB 1464, to assess the range of activity of the extract. The same *P. syringae* strain was also utilized for the in-planta experiments. The *Chromobacterium violaceum* strain NCTC 9757 for the quorum sensing test was provided by the Spanish Type Culture Collection (CECT), along with *B. subtilis* NCTC 8236 and *P. syringae* NCPPB 1464 pv. tomato. Bacterial strains were cultivated on EBS medium (in g per L: casein peptone, 5; D-glucose, 5; meat extract, 1; yeast extract, 1; 50 mM Hepes, 11.9) at 30 °C in the dark for 24 h to obtain sufficient inocula. Colony-forming unit (CFU) concentrations were calculated using a Neubauer chamber. For the greenhouse experiments, commercial seeds of *Solanum lycopersicum* (Cultivar Marmande) were utilized.

### 2.2. Filtrate and Extract Obtention

Two 5 mm agar discs were excised from an actively growing colony (7-day-old) of the endophyte and placed into each of three 500 mL Erlenmeyer flasks containing 250 mL of yeast malt broth (YMB, in g per L: yeast extract: 6; malt extract: 10; D-glucose: 6; pH adjusted to 6.3). The flasks were then incubated in a thermoshaker (Orbital Shaker Incubator COMECTA 1102, Barcelona, Spain) at 23 °C and 140 rpm. Two days after the total consumption of glucose in the medium, the fungal culture was filtered using sterile paper discs (pore = 0.2 μm) to separate the mycelium from the liquid filtrate containing the secondary metabolites [33]. A portion of this filtrate was subjected to an extraction process following the protocol described by Halecker et al. [34], mixing the filtrate with an equal volume of ethyl acetate and shaking vigorously for 2 min. The resulting mixture was then poured into a separatory funnel to allow the two phases to separate. To remove any water residues, a small amount of sodium sulfate was added to the organic phase, which was subsequently filtered through 0.16 mm filters (MN 615 ¼). The resulting sample was evaporated using a rotary evaporator to remove the ethyl acetate (Hei-Vap ML/G1, Thermo Fisher Scientific, Waltham, MA, USA). The solid residue was then resuspended in methanol for the in vitro tests, in dimethyl sulfoxide (DMSO, Panreac, Barcelona, Spain) for the quorum sensing activity tests, and in water for the in-planta tests.

### 2.3. Effect of the Fungal Filtrate on the Bacterial Growth In Vitro (Disc Diffusion Assay)

The antimicrobial activity was initially assessed using the disc diffusion method [35]. For this assay, plates (3 replicates per bacterial species) were divided into quadrants. Then, 18 mL of potato dextrose agar (PDA, VWR Chemicals, Pennsylvania, PA, USA, 39 g L$^{-1}$) was poured into each plate. Before solidification, 2 mL per plate of bacterial solution was added to achieve a concentration of $5 \times 10^4$ CFU in the total 20 mL and gently shaken for homogenization. After solidification, a sterile 6 mm diameter filter paper containing 20 μL of the filtrate was placed in the center of each plate. Penicillin, at a concentration of 1.5 mg mL$^{-1}$, was used as the positive control, while sterile YMB served as the negative control. The plates were then incubated at 30 °C in the dark. In each quadrant, the radius

of inhibition produced by the filtrate was measured every 12 h up to 72 h. Based on these measurements, the effective radius of action of the filtrate, where no bacterial growth occurred, was calculated as the average across each quadrant.

### 2.4. Effect of the Extract on the Quorum Sensing Activity

The efficacy of the *A. leptinellae* 138 extract in inhibiting bacterial quorum sensing activity was tested based on its ability to prevent the production of violacein by *Chromobacterium violaceum* NCTC 9757. The agar diffusion method was employed for this purpose [36,37]. Initially, 50 μL of the organism *C. violaceum* NCTC 9757, cultivated overnight at 30 °C with agitation, were added at a concentration of $10^8$ CFU to 5 mL of liquid Luria-Bertani (LB, in g per L: casein peptone, 5; yeast extract, 5; NaCl, 5; agar, 15; pH adjusted to 7.22) agar at 40 °C. Before solidification, 15 mL of the mixture was poured into Petri dishes. Once the medium solidified, 6 mm diameter wells were made and filled with 250 μg of the extract dissolved in 500 μL of DMSO. The plates were then incubated in the dark at 30 °C for 12 h. DMSO was used as the negative control, and 20 μg of the compound (Z-)-4-bromo-5-(bromomethylene)-2(5H)-furanone (Sigma Aldrich, San Luis, MO, USA), a known quorum sensing inhibitor [38], was used as the positive control. Inhibition, indicated by an opaque halo around the disc, was estimated using the equation R2–R1 (in mm), where R1 corresponds to the radius of inhibition of bacterial growth (transparent zone), and R2 includes R1 + the zone where violacein production was inhibited but not the microorganism growth (opaque white zone).

### 2.5. In Vitro Minimum Inhibitory Concentration of the Extract

The minimum inhibitory concentration (MIC) of the fungal extract against *Bacillus subtilis* NCTC 8236 and *Pseudomonas syringae* NCPPB 1464 was determined using a broth microdilution assay in 96-well plates, following the protocol described by Halecker et al. [34]. A freshly prepared solution ($6.7 \times 10^5$ CFU mL$^{-1}$) of each bacterial species (130 μL) was added to each well, followed by 20 μL (300 μg mL$^{-1}$) of the extract in the first row. Subsequently, a 50% serial dilution was performed for the subsequent rows. To ensure the validity of the test, a positive control of 1.5 mg mL$^{-1}$ of penicillin and a negative control of 20 μL of methanol were included. The plates were then incubated at 30 °C and 600 rpm for 48 h before the results were evaluated. At the end of this period, growth inhibition was qualitatively assessed based on the turbidity or clarity of the culture medium. The test was performed in triplicate.

### 2.6. In Planta Assays

To assess the efficacy of the extract in protecting tomato plants against *Pseudomonas syringae* NCPPB 1464, two greenhouse experiments were conducted: (i) a mycopriming assay based on the application of the fungal extract to seeds, and (ii) a post-emergence assay, where extracts were applied to seedlings. Before both experiments, tomato seeds were surface-disinfected by immersion in 70% ethanol for one minute, followed by subsequent immersion in a 2% sodium hypochlorite solution for one minute. Afterward, seeds were rinsed three times with sterile distilled water to remove any residual disinfectant [39].

In the mycopriming assay, the disinfected seeds were immersed in the fungal extract (3 mg of extract per mL of sterilized distilled water) for 6 h to ensure optimum uptake of metabolites. Seeds immersed for the same period in sterile distilled water, the solvent used for the extract, served as controls. Sixty seeds per treatment were then individually sown in separate plastic pots (7 × 7 × 6 cm) containing a mixture of substrate and perlite, with a pH of 7.00 ± 0.50, EC of 1.50 ± 0.10 dS m$^{-1}$, organic matter = 60.0 ± 2.0%, N = 1.29 ± 0.08%, $P_2O_5$ = 0.58 ± 0.05%, and $K_2O$ = 1.25 ± 0.10%. Before sowing, half of the pots were inoculated with a solution containing the pathogen (100 mL solution per L soil, adjusted to $2 \times 10^4$ CFU mL$^{-1}$ of *P. syringae* NCPPB 1464), while the other half received the same volume of a pathogen-free solution. The pots were then placed in a greenhouse for 15 days and watered every 2–3 days until reaching field capacity. Germination was monitored daily.

At the end of the 15-day period, five plants from each treatment combination were randomly harvested. Shoot and root elongation, as well as the number of roots, were measured for each plant. The vigor index of each treatment was calculated by multiplying the respective germination rate (in %) at the end of the test by the shoot length of the corresponding samples. To ensure the validity of the test, this assay was performed simultaneously in triplicate, with each set of 60 seeds considered as a repetition. The experiment took place from 18 February to 4 March 2020.

In the post-emergence test, an additional 60 disinfected seeds were individually sown in pots containing the same substrate mixture, placed in the greenhouse, and watered every 2–3 days until reaching field capacity. After one month, half of the pots received an application of 5 mL of the pathogen solution ($2 \times 10^4$ CFU mL$^{-1}$ of *P. syringae* NCPPB 1464) per plant. Twelve hours later, each plant was treated with the endophyte by spraying a dose of 1 mL of the extract (3 mg mL$^{-1}$). A control treatment with 1 mL of sterilized distilled water was included. To assess treatment efficacy, disease severity was measured weekly for one month after application, based on the percentage of damaged leaves, with three leaves per plant randomly selected [40]. To quantify disease progression in each pot, the area under the disease progression curve (AUDPC) was calculated by summing the areas of the corresponding trapezoids. Each period between two consecutive measurements was considered as a unit. After the final measurement, five plants from each treatment were randomly selected for laboratory analysis, where the number and lengths of roots, as well as shoot length, were measured. Herbage and root dry weights were also determined after oven drying the samples at 60 °C until constant weight. To ensure the validity of the results, the assay was conducted simultaneously in duplicate. This experiment was carried out from 18 February to April 30, 2020. The temperature and humidity conditions recorded during both experiments mycopriming and post-emergence are shown in Figure 1.

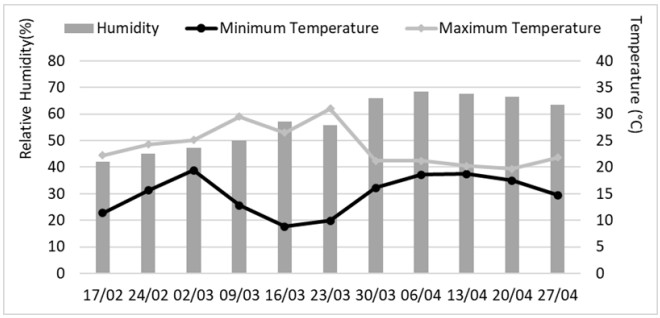

**Figure 1.** Temperature and relative humidity in the greenhouse during the greenhouse assays. For each date, the average value is shown for each week starting from the specified day.

## 2.7. Evaluation of Other Protective Traits in Alternaria leptinellae E138

The potential of *A. leptinellae* E138 to produce substances directly or indirectly involved in plant protection was assessed using tree key traits: (i) production of phytohormone-like substances, (ii) antioxidant activity and synthesis of phenolic compounds, and (iii) nutrient mobilization, including phosphate solubilization capacity, siderophore production, and ammonia production.

### 2.7.1. Estimation of Phytohormone-like Substances Production

To determine the production of phytohormone-like substances, we assessed both auxin-like (indoleacetic acid, IAA) and gibberellin-like (gibberellic acid equivalents, GAE) substances in the fungal filtrate. All samples were analyzed in triplicate, and the results were expressed as milligrams of compound (i.e., IAA or GAE, respectively) per milliliter of fungal filtrate. For IAA quantification in the fungal filtrate, a colorimetric test was employed [41]. Salkowski reagent was mixed with the fungal filtrate, and after incubation in the dark for 30 min, the absorbance was measured at 530 nm using a spectrophotometer (JP Selecta UV 3100). The concentration of IAA was determined using a regression equation

based on a standard curve of pure IAA (Sigma Aldrich). Additionally, the endophyte was cultured in YMB medium supplemented with L-tryptophan (5 mM), to enhance IAA production [42]. For GAE quantification, a colorimetric method was also employed. Firstly, 15 mL of the fungal filtrate was mixed with 2 mL of zinc acetate (21.9%) and 2 mL of potassium ferrocyanide (10.6%) sequentially at two-minute intervals. The mixture was then centrifuged at 2000 rpm for 15 min. The supernatant was mixed with 5 mL of HCl (30%) and incubated at 20 °C for 75 min. Absorbance was measured at 254 nm, and the results were compared with a calibration curve for gibberellic acid (GA3, Sigma Aldrich) [43]. All samples were analyzed in triplicate.

### 2.7.2. Determination of the Antioxidant Activity

To assess the antioxidant activity, the fungal extract of *A. leptinellae* E138 was subjected to a DPPH (1,1-diphenyl-2-picrylhydrazyl) assay [44] with slight modifications. A volume of 0.1 mL of the fungal extract at a concentration of 3 mg mL$^{-1}$ was mixed with 2.9 mL of 0.004% aqueous DPPH, and the mixture was incubated in the dark at 25 °C for 30 min. Absorbance was then measured at 517 nm using a spectrophotometer (JP Selecta UV 3100). The DPPH scavenging capacity was calculated using the following formula: % Radical scavenging = (Absorbance Control − Absorbance Sample)/Absorbance Control × 100. All samples were analyzed in triplicate.

### 2.7.3. Determination of Total Polyphenol Content (TPC)

The Folin–Ciocalteau method was employed to determine the TPC in the fungal extract of *A. leptinellae* E138 [45]. For this analysis, 1 mL of the fungal extract (1 mg mL$^{-1}$) was mixed with 500 μL of 50% aqueous Folin–Ciocalteau reagent, 1.5 mL of 20% aqueous $Na_2CO_3$, and 2 mL of distilled water. The mixture was then incubated at room temperature in the dark for 30 min, and the absorbance was measured at 765 nm. The polyphenol content was expressed as milligrams of gallic acid equivalents per gram of fungal extract. All samples were analyzed in triplicate.

### 2.7.4. Determination of Nutrient Mobilization

Three parameters were evaluated in *A. leptinellae* E138 for nutrient mobilization: phosphate solubilization capacity, siderophore production, and ammonia production.

**Phosphate solubilization.** For the qualitative assessment, an actively growing plug of *A. leptinellae* E138 was placed in a Petri dish containing the National Botanical Research Institute's phosphate growth medium (NBRIP) amended with 1.5% agar [46,47]. Plates were incubated at 27 °C for 7 days and then examined for a clear halo around the colony. The diameter of this clear zone was measured, and the solubilization capacity of the endophyte was estimated using the following formula: Solubilization Index (%) = (Colony diameter + Clear zone diameter)/(Colony diameter). Three replicates were performed. For quantitative assessment, a piece of active mycelium was inoculated into a flask containing 50 mL of agar-free NBRIP medium and incubated on a shaker at 27 °C and 140 rpm for 15 days. During this period, three aliquots of 10 mL each were taken on days 5, 10, and 15 to assess the pH variation and phosphorus content [48]. The results were compared with a calibration curve of $Ca_3(PO_4)_2$. In both assays, a blank was introduced using a piece of PDA medium without any fungal inoculation and the experiments were conducted in triplicate.

**Siderophore production.** Siderophore production was estimated using a modified Chrome Azurol S (CAS) universal assay [49]. An actively growing plug of *A. leptinellae* E138 was placed in the center of Petri dishes containing 20 mL of Minimal Medium 9 (MM9) and incubated in a growth chamber at 27 °C for 7 days. After incubation, a modified CAS solution (60.5 mg of CAS, 72.9 mg of HDTMA, 30.24 g of PIPES, and $FeCl_3 \cdot 6H_2O$ in 10 mL 10 mM HCl and agarose 0.9%, *w/v*) was added to each Petri dish. After 15 min, a color change was observed in the medium surrounding the siderophore-producing microorganisms. The size of the halo was measured, and the siderophore production was calculated using the same formula as used for the phosphate solubilization.

Additionally, the protocol was repeated with a non-deferrated medium to confirm that siderophore production was not induced under normal conditions. For both assays, a blank was introduced using a plug of uninoculated PDA medium, and three replicates were performed.

**Ammonia production.** Qualitative assessment of ammonia production was performed by growing *A. leptinellae* E138 in peptone water at 28 °C for 72 h. After incubation, Nessler's reagent was added, and the color change indicated the degree of ammonia production. A pale-yellow color indicated minimal ammonia production, while a deep yellow to brown color indicated maximum ammonia production [50]. The test was also performed in triplicate.

*2.8. Mass Spectometry Analysis of the Fungal Extract*

For a tentative identification of the metabolites present in the fungal extract, mass spectrometry was performed using an Agilent 6520 Accurate Mass Q-TOf LC/MS system (Agilent, Santa Clara, CA, USA) with an electrospray ionization interface in positive ion mode. The operating parameters were as follows: capillary voltage, 3500 V; fragmenter, 100 V; nebulizer pressure, 35 psig; drying gas temperature, 300 °C; acquisition range 150–800 $m/z$. Nitrogen was employed as the drying gas at a flow rate of 12.0 L min$^{-1}$. The system also included a diode array detector operating in the 280 to 350 nm range with a 2 nm step. Samples were eluted on an Agilent Zorbax eEclipse Plus C18 Rapid Resolution column (4.6 × 100 mm, 3.5 μm) maintained at 30 °C. The mobile phase consisted of 0.1% formic acid in ultrapure water (obtained from Millipore Integral-5 purification system) (solvent A) and 0.1% formic acid in acetonitrile (solvent B). A gradient elution was applied as follows: 0–10% B (0 min); 10–100% B (30 min); 100% B isocratic mode (10 min); and for column reconditioning, 100–10% B (1 min) and 10% B (7 min). Both formic acid and acetonitrile were of LC/MS grade. The flow rate was set to 0.30 mL min$^{-1}$, and the injection volume was 1 μL. By examining the information obtained from the HPLC analysis, including observed mass/charge relationships and proposed formulae, compounds detected by the Q-TOf LC/MS system were tentatively assigned through a literature search.

*2.9. Statistical Analysis*

One-way ANOVAs and Mixed-effects ANOVAs were used to assess the effect of the filtrate or the extract on bacterial growth and response variables in the greenhouse assays, respectively. The factors considered were endophyte treatment, pathogen inoculation, and their interaction. A Fisher's protected least significant difference (LSD) test at $p \leq 0.05$ was applied for post hoc analyses. Prior to the analysis, assumptions of normal distribution and homoscedasticity were ensured by Shapiro–Wilk and Levene's tests, respectively. All analyses were performed with the Statistix v. 8.10 package (Analytical Software, Tallahassee, FL, USA).

**3. Results**

*3.1. Effect of the Filtrate and the Extract of Alternaria leptinellae E138 on the Bacterial Growth and Quorum Sensing Activity In Vitro*

The summary of the one-way ANOVA summary, presented in Table 1, indicates a significant decrease ($p < 0.05$) caused by the endophyte *A. leptinellae* E138 filtrate on the in vitro growth of *Bacillus subtilis* NCTC 8236 and *Pseudomonas syringae* NCPPB 1464, as well as on the quorum sensing activity of *Chromobacterium violaceum* NCTC 9757. In the disc diffusion assay, *A. leptinellae* E138 produced inhibition zones of 0.95 mm and 0.39 mm against each bacterial species, respectively. These values represent 50% and 57.4% of the respective inhibitory effect of penicillin, used as a reference control. The blank (sterilized YM broth medium) showed no inhibitory effect, indicating that the observed inhibition was due to the compounds present in the filtrate. In the evaluation of quorum sensing activity (QSA) using *C. violaceum* NCTC 9757, the *A. leptinellae* E138 extract exhibited an

inhibitory effect with an inhibition zone value of 1.38 mm, suggesting a potential role in disrupting bacterial communication mechanisms. In this case, it accounted for 27.6% of the inhibitory effect of the compound (Z-)-4-bromo-5-(bromomethylene)-2(5H)-furanone, used as a control. Positive values for the minimum inhibitory concentration (MIC) were obtained. Thus, the inhibition of *B. subtilis* NCTC 8236 or *P. syringae* NCPPB 1464 was achieved with a concentration of the extract of 125 µg mL$^{-1}$ and 300 µg mL$^{-1}$, respectively.

**Table 1.** Effect of *Alternatia leptinellae* E138 filtrates and extracts on the bacterial growth and the quorum sensing activity (QSA).

| | Disc Diffusion Assay (mm) | | QSA Inhibition (mm) | MIC (µg Extract mL$^{-1}$) | |
| --- | --- | --- | --- | --- | --- |
| | *Bacillus subtilis* | *Pseudomonas syringae* | | *Bacillus subtilis* | *Pseudomonas syringae* |
| E138 | 0.95 ± 0.03 b | 0.39 ± 0.04 b | 1.38 ± 0.13 b | 125.00 ± 25.00 | 300.00 ± 0.00 |
| Blank [1] | 0.00 ± 0.00 c | 0.00 ± 0.00 c | 0.00 ± 0.00 c | 0.00 ± 0.00 | 0.00 ± 0.00 |
| Control [2] | 1.90 ± 0.06 a | 0.68 ± 0.04 a | 5.00 ± 0.38 a | 1.24 ± 0.00 | 1.24 ± 0.00 |
| df | 2 | 2 | 2 | - | - |
| Endophyte | 689.18 *** | 117.72 *** | 127.25 *** | - | - |

QSA on *Chromobacterium violaceum* NCTC 9757. Values (estimated by the radius of the inhibition zone, in the disc diffusion assay) are expressed as mean ± error standard ($n = 4$). Different letters for each column indicate significant differences according to LSD (least significant difference) test at $\alpha = 0.05$. [1] Blank: sterilized YM broth medium for the disc diffusion assay, and methanol for the MIC assay. [2] Control: penicillin (1.5 mg mL$^{-1}$) and (Z-)-4-bromo-5-(bromomethylene)-2(5H)-furanone (20 µg), in the disc diffusion assay and QSA, respectively. A summary of the one-way ANOVAs in the disc diffusion assays is shown at the bottom, indicating the degree of freedom (df), *F*-value and the level of significance (*** $p \leq 0.001$).

*3.2. Effect of Mycopriming with A. leptinellae E138 Extract on the Control of Pseudomonas syringae NCPPB 1464 in Tomato Plants under Greenhouse Conditions*

Regarding the germination rate of *S. lycopersicum*, it was significantly affected by the treatment with the endophytic extract and its interaction with the presence or absence of the pathogen *P. syringae* NCPPB 1464 for each day of measurement (from the fourth day until the fifteenth day after sowing), except on the sixth and seventh days (Table 2). This variability in the nature of the relationship between the endophyte extract and the pathogen, manifested in the different *F*- and *p*-values among days, tended to stabilize from day 9 onwards.

**Table 2.** ANOVAs showing the effect of *Alternaria leptinellae* E138 extract, *Pseudomonas syringae* NCPPB 1464, and their interaction on the germination rate of *Solanum lycopersicum* seeds.

| | Day 4 | Day 5 | Day 6 | Day 7 | Day 8 | Day 9 |
| --- | --- | --- | --- | --- | --- | --- |
| df | 1 | 1 | 1 | 1 | 1 | 1 |
| Endophyte (E) | 8.00 * | 25.48 ** | 180.44 *** | 112.45 *** | 98.01 *** | 484.99 *** |
| Pathogen (P) | 11.99 | 21.07 * | 3.57 | 12 | 2.28 | 4.92 |
| E × P | 16.00 * | 48.02 * | 3.00 | 3.00 | 15.93* | 99.91 ** |
| | Day 10 | Day 11 | Day 12 | Day 13 | Day 14 | Day 15 |
| df | 1 | 1 | 1 | 1 | 1 | 1 |
| Endophyte (E) | 264.66 *** | 220.67 *** | 242.20 *** | 72.09 ** | 72.09 ** | 72.09 ** |
| Pathogen (P) | 7 | 6.26 | 12.01 | 12.01 | 12.01 | 12.01 |
| E × P | 121.45 ** | 360.00 ** | 64.07 * | 100.36 ** | 100.36 ** | 100.36 ** |

The degrees of freedom (df), the *F*-values, and the levels of significance (* $p \leq 0.05$; ** $p \leq 0.01$; *** $p \leq 0.001$) are shown for each day of measurement.

Analysis of germination rates for tomato seeds, as presented in Figure 2, revealed different patterns depending on the presence or absence of *P. syringae* NCPPB 1464 and the application of *A. leptinellae* extract E138 compared to the control group. In the absence of *P. syringae* NCPPB 1464, the control group without extract (Control−) exhibited a steady increase in germination rates from day 4 to day 9, and then stabilized until day 15, reaching a germination rate of around 71% at the end. In contrast, in presence of the pathogen (Control+), there was a significant decrease in germination rates, reaching a maximum (only 53.33%) on day 11, demonstrating the pathological nature of the strain used. The application of the *A. leptinellae* extract (138−) increased the germination rate compared to the controls (Control−) and also reverted the negative impact of the pathogen when

it was also inoculated (138+). Thus, regardless of the presence or absence of *P. syringae* NCPPB 1464, the germination rate reached maximum values of 86–91% by day 13 when the endophyte extract was applied. This result demonstrates not only the lack of toxicity of the endophyte extract on tomato seeds, at least in terms of their germination rate, but also a clear biocontrol and plant-growth-promoting activity capable of reverting the negative effects of *P. syringae* NCPPB 1464.

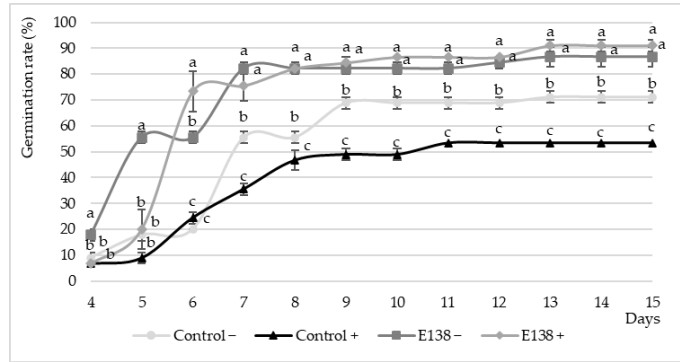

**Figure 2.** Effect of endophytic extract (E138) on the germination percentage of *Solanum lycopersicum* seeds inoculated (+) and not inoculated (−) with *Pseudomonas syringae* NCPPB 1464 from 4 to 15 days after sowing (12 measurements). Values are expressed as mean ($n$ = 3) ± standard error (error bars). In each day of measurement, different letters indicate significant differences between treatments according to the LSD (least significant difference) test at α = 0.05.

In relation to the growth parameters of tomato plants under greenhouse conditions, both the treatment with *A. leptinellae* E138 extract and inoculation with *P. syringae* NCPPB 1464 significantly affected shoot and root lengths, the number of roots, and the vigor index of seedlings. The interaction between these factors demonstrated significant influence ($p < 0.05$) on shoot and root lengths and vigor index (Table 3). The presence of the pathogen led to a significant decrease in root length and vigor index compared to the control group (root length: 1.43 cm vs. 1.85 cm and vigor index: 466.64 vs. 661.32). Application of the E138 extract not only reverted the negative effect of the pathogen but also resulted in increased parameters compared to the controls (root length: 2.58 cm and vigor index: 924.77). This increase can be attributed to both genuine biocontrol activity and clear plant-growth-promoting effects, as evidenced by superior growth parameters in the absence of the pathogen (root length: 3.48 cm and vigor index: 951.17; Table 3). In this latter case, there was also a significant increase in the number of roots (2.25 vs. 1.00). These increases were 88.11%, 125%, and 43.83%, respectively, compared to seedlings in the control treatment growing in the absence of the pathogen.

**Table 3.** Growth parameters of tomato plants (shoot length, root length, number of roots, and vigor index) as affected by treatment of seeds with the *Alternaria leptinellae* E138 extract, inoculation with *Pseudomonas syringae* NCPPB 1464, and their interaction under greenhouse conditions.

| Endophyte | Pathogen | Shoot Length (cm) | Root Length (cm) | Number of Roots | Vigor Index |
|---|---|---|---|---|---|
| Control | No | 9.30 ± 0.18 | 1.85 ± 0.05 c | 1.00 ± 0.00 b | 661.32 ± 12.65 b |
| | Yes | 8.75 ± 0.25 | 1.43 ± 0.08 d | 1.00 ± 0.00 b | 466.64 ± 13.33 c |
| E138 | No | 10.98 ± 0.24 | 3.48 ± 0.05 a | 2.25 ± 0.25 a | 951.17 ± 20.74 a |
| | Yes | 10.15 ± 0.15 | 2.58 ± 0.05 b | 1.00 ± 0.00 b | 924.77 ± 13.67 a |
| df | | 1 | 1 | 1 | 1 |
| Endophyte (E) | | 46.55 *** | 443.53 *** | 25.00 ** | 613.66 *** |
| Pathoten (P) | | 10.56 * | 238.47 *** | 25.00 * | 42.34 ** |
| E × P | | 0.68 | 21.24 * | 25.00 * | 33.64 * |

Values are expressed as mean ($n$ = 5) ± standard error. For each parameter, different letters (if any) indicate significant differences according to LSD (least significant difference) test at α = 0.05. A summary of the ANOVAs is also given at the bottom of the table. The degrees of freedom (df), the *F*-values, and the levels of significance (* $p \leq 0.05$; ** $p \leq 0.01$; *** $p \leq 0.001$) are given for each parameter.

### 3.3. Effect of the Post-Emergence Application of A. leptinellae E138 Extract on the Control of Pseudomonas syringae NCPPB 1464 in Tomato Plants under Greenhouse Conditions

Regarding post-emergence application, the ANOVAs revealed that both the application of *A. leptinellae* extract E138 and the inoculation with *P. syringae* NCPPB 1464, had a significant influence on all the growth parameters studied (Table 4). The pathogenicity of the *P. syringae* NCPPB 1464 strain used was demonstrated by its inoculation resulting in a significant decrease in all parameters compared to the controls (shoot length: 4.88 cm vs. 26.64 cm, root length: 4.66 cm vs. 16.40 cm, roots per plant: 8.40 vs. 20.20; Table 4). Once again, the application of the endophyte extract not only reverted the negative effects of the pathogen (resulting in a 2.34-fold, 1.42-fold, and 2-fold increase in shoot, root, and total dry matter weights, respectively, compared to the plants inoculated with the pathogen), but also enhanced these growth parameters compared to the non-inoculated controls (with increases of 35.66% and 88.12% for shoot length and root length, respectively). Finally, the AUDPC showed a partial reversion of the symptoms exhibited by plants inoculated with *P. syringae* NCPPB 1464, when treated with the extracts, resulting in a 41.10% reduction in disease severity (Table 4). The non-inoculated plants treated with the endophyte extract did not exhibit any disease symptoms, further demonstrating its non-phytotoxic nature in tomato plants.

**Table 4.** Effect of the post-emergence application of the extract of *Alternaria leptinellae* E138, the inoculation with the pathogen *Pseudomonas syringae* NCPPB 1464, and their interaction on the different growth traits (shoot and root length; number of roots per plant and number of plants per pot; shoot, root, and total dry weight), and on the disease severity (estimated though the area under the disease progression curve; AUDPC) in tomato plants under greenhouse conditions.

| Endophyte | Pathogen Presence | Shoot Length (cm) | Root Length (cm) | Number of Roots | AUDPC |
|---|---|---|---|---|---|
| Control | No | 26.64 ± 1.71 b | 16.40 ± 0.63 b | 20.20 ± 0.56 b | 0.00 ± 0.00 c |
| | Yes | 4.88 ± 2.99 c | 4.64 ± 2.85 c | 8.40 ± 5.15 c | 66.50 ± 1.29 a |
| E138 | No | 36.14 ± 2.10 a | 21.54 ± 0.64 a | 38.00 ± 0.71 a | 0.00 ± 0.00 c |
| | Yes | 38.00 ± 1.24 a | 16.16 ± 0.11 b | 42.60 ± 0.93 a | 39.20 ± 3.06 b |
| | | Shoot dry matter weight (g) | Root dry matter weight (g) | Total dry matter weight (g) | |
| Control | No | 173.05 ± 7.47 b | 74.35 ± 3.60 b | 247.4 ± 9.16 b | |
| | Yes | 37.72 ± 23.85 c | 9.638 ± 5.91 c | 47.358 ± 29.66 c | |
| E138 | No | 578.35 ± 32.17 a | 180.65 ± 14.47 a | 759.00 ± 39.37 a | |
| | Yes | 581.05 ± 26.52 a | 163.375 ± 7.38 a | 744.425 ± 19.92 a | |
| | df | Shoot Length (cm) | Root Length (cm) | Number of roots | AUDPC |
| Endophyte (E) | 1 | 229.50 *** | 43.47 *** | 93.5 *** | 67.36 *** |
| Pathogen (P) | 1 | 11.60 * | 21.06 * | 1.59 | 2205.82 *** |
| E × P | 1 | 25.91 ** | 4.54 * | 11.87 * | 43.67 * |
| Source | df | Shoot dry matter weight (g) | Root dry matter weight (g) | Total dry matter weight (g) | |
| Endophyte (E) | 1 | 276.01 *** | 273.98 *** | 412.61 *** | |
| Pathogen (P) | 1 | 12.44 * | 11.92 * | 17.98 * | |
| E × P | 1 | 12.58 * | 11.89 * | 17.2 * | |

Values are expressed as mean ($n$ = 5) ± standard error. For each parameter, different letters indicate significant differences according to LSD (least significant difference) test at α = 0.05. A summary of the ANOVAs is also given at the bottom of the table. The degrees of freedom (df), the *F*-values, and the levels of significance (* $p \leq 0.05$; ** $p \leq 0.01$; *** $p \leq 0.001$) are given for each factor.

### 3.4. Traits of A. leptinellae E138 Directly or Indirectly Related to Its Biocontrol Activity

In terms of phytohormone-like production, the results indicate that the endophyte E138 was capable of producing indole-acetic acid (IAA), particularly when the culture media were supplemented with tryptophan (IAA+, resulting in a threefold increase), and gibberellins (GA3) (Table 5). Regarding antioxidant activity, although *A. leptinellae* E138 produced a total polyphenol content of 68.02 mg GAE (gallic acid equivalent) per gram of extract and 43.35 mg of quercetin equivalent per gram of extract, it exhibited minimal DPPH scavenging activity, with a result of only 1.47% (Table 5). The endophyte also showed a positive potential for ammonia synthesis and phosphate solubilization in qualitative tests.

**Table 5.** Traits of *Alternaria leptinellae* E138 related to its biocontrol activity (phytohormone-like production, antioxidant activity, and nutrient mobilization capacity).

| | Trait | |
|---|---|---|
| **Group** | **Activity** | **Mean $\pm$ SE** |
| Phytohormone | IAA ($\mu$g mL$^{-1}$) | 6.10 $\pm$ 0.05 |
| | IAA+ ($\mu$g mL$^{-1}$) | 19.22 $\pm$ 0.20 |
| | GA$_3$ ($\mu$g mL$^{-1}$) | 442.88 $\pm$ 0.36 |
| Antioxidant activity | DPPH (%) | 1.47 $\pm$ 0.61 |
| | TPC (mg GAE g$^{-1}$) | 68.02 $\pm$ 4.46 |
| Nutrient mobilization | Siderophore | $-$ |
| | P solubilization | + |
| | Ammonia production | ++ |

IAA, indole-acetic acid in the filtrate; IAA+, indole-acetic acid in the filtrate with culture media supplemented with L-tryptophan; GA3, gibberellic Acid in the filtrate; DPPH, % of scavenging of DPPH in the extract; TPC, total polyphenol content in the extract. For the parameters related to nutrient mobilization, the positive or negative potential to produce each type of compounds is shown ($-$ no production; + production; ++ high production). Quantitative data are expressed as mean $\pm$ standard error ($n$ = 3).

Furthermore, in the quantitative assay, the endophyte E138 demonstrated significant phosphorus solubilization capabilities, as evidenced by the increased percentage of solubilized P Olsen after 5, 10, and 15 days. The sustained increase in solubilization over time indicates the efficiency of E138 in rendering phosphorus more available to plants without acidifying the environment (Table 6).

**Table 6.** Quantitative analysis of the phosphorus solubilization in vitro by *Alternaria leptinellae* E138, showing the percentage of P Olsen solubilized after 5, 10 and 15 days, together with the pH measured at each moment.

| | **P Solubilization Day 5 (%)** | **P Solubilization Day 10 (%)** | **P Solubilization Day 15 (%)** |
|---|---|---|---|
| E138 | 67.60 $\pm$ 0.34 a | 68.34 $\pm$ 0.64 a | 69.67 $\pm$ 0.41 a |
| Control | 0.47 $\pm$ 0.03 b | 0.08 $\pm$ 0.10 b | 0.10 $\pm$ 0.13 b |
| df | 1 | 1 | 1 |
| Endophyte | 8502.89 *** | 16,798.50 *** | 37,195.30 *** |
| | **pH day 5** | **pH day 10** | **pH day 15** |
| E138 | 6.51 $\pm$ 0.08 | 6.70 $\pm$ 0.05 a | 6.68 $\pm$ 0.02 a |
| Control | 6.21 $\pm$ 0.05 | 6.12 $\pm$ 0.04 b | 5.80 $\pm$ 0.00 b |
| df | 1 | 1 | 1 |
| Endophyte | 5.11 | 24.14 ** | 91.58 *** |

Values are expressed as mean ($n$ = 3) $\pm$ standard error. A summary of ANOVAs is also given at the bottom of the table. Different letters for each column indicate significant differences according to LSD (least significant difference) test at $\alpha$ = 0.05. The degrees of freedom (df), the *F*-values, and the levels of significance (** $p \leq 0.01$; *** $p \leq 0.001$) are given for each parameter.

### 3.5. Tentative Identification of Metabolites by Mass Spectometry

As a preliminary attempt to identify the metabolites responsible for the observed plant protection effects, we analyzed the compounds present in the fungal extract obtained through mass spectrometry analysis. Compounds were tentatively assigned by examining mass/charge relationships and proposed formulae through a literature search. Mass spectrometry graphs associated with each compound are displayed in Supplementary Figures (Figures S1–S4). Table 7 provides a comprehensive overview of the compounds identified in the methanolic extracts of *A. leptinellae* E138, detailing their molecular characteristics and activities. As a result, the following compounds were proposed: altechromone A, maculosin, ciclo(L-Phe-L-Pro), and phomopsinone A. All of these compounds were found to have activities consistent with the results obtained in the different experiments con-

ducted in this study, ranging from plant growth promotion to antimicrobial and antifungal activities. However, these preliminary results require further research to unequivocally confirm their identification.

**Table 7.** Peak assignment for methanolic extracts of *Alternaria leptinellae* E138.

| N° | Proposed Formula | Rt (min) | Obs. *m/z* | Proposed Compound | Activity | Reference |
|---|---|---|---|---|---|---|
| **Cpd01** | $C_{11}H_{10}O_3$ | 5.344 | 190.0623 | Altechromone A | Plant growth promotion, antimicrobial activity (biofilm inhibition) | [51,52] |
| **Cpd02** | $C_{14}H_{16}N_2O_3$ | 8.836 | 260.1155 | Maculosin | Antibacterial and antioxidant activity | [53] |
| **Cpd03** | $C_{14}H_{16}N_2O_2$ | 12.979 | 244.1205 | Ciclo(L-Phe-L-Pro) | Influence on QSA | [54] |
| **Cpd04** | $C_{12}H_{16}O_4$ | 15.242 | 224.1042 | Phomopsinone A | Antifungal activity | [55,56] |

Rt: retention time; Obs *m/z*: observed mass/charge relationship; QSA: quorum sensing activity.

## 4. Discussion

The results of our study explore the potential of *A. leptinellae* E138 as a biocontrol agent against *Pseudomonas syringae* in tomato plants, emphasizing its multifaceted action as an antimicrobial and a plant growth promoter. In this study, we focused on the direct use of the filtrates and extracts produced by the endophyte, as their effectiveness has already been demonstrated in other cases, overcoming the drawbacks associated with using the living organism [57,58]. Results regarding the effect of *A. leptinellae* E138 filtrates on bacterial growth in vitro demonstrate significant biocontrol potential against *B. subtilis* NCTC 8236 and *P. syringae* NCPPB 1464. This was supported by MIC tests and the evaluation of quorum sensing activity (QSA), wherein the endophyte extracts inhibited the growth of both bacteria and disrupted QSA in *C. violaceum* NCTC 9757.

The observed inhibition zones in the disc diffusion assay, although slightly smaller than the reference control (penicillin), underscore the potential of *A. leptinellae* E138 as a source of antimicrobial compounds that may prove valuable in biocontrol strategies against *P. syringae*. Furthermore, the effect of the endophyte on QSA adds another dimension to its biocontrol mechanisms. Disrupting bacterial communication mechanisms may reduce the coordination of pathogenic activities, providing an additional layer of defense against *P. syringae* [59]. This is consistent with the findings of Adonizio et al. [60], who demonstrated the potential of inhibiting the quorum sensing activity of a *P. aeruginosa* bacterial colony to diminish its virulence factor. The MIC values obtained evidenced the efficacy of the *A. leptinellae* E138 extract against *P. syringae* NCPPB 1464, given the low concentration required to inhibit the pathogen (300 $\mu$g mL$^{-1}$). This highlights the practical feasibility of incorporating *A. leptinellae* E138 extract into biocontrol strategies. This agrees with the findings of Singh et al. [61], who demonstrated the anti-biofilm and anti-quorum sensing activity of microbial extracts against *Pseudomonas* sp. Understanding the MIC values is crucial for determining appropriate concentrations in field applications to ensure optimal efficacy against target pathogens.

This antimicrobial activity observed in *A. leptinellae* E138 filtrates and extracts may be attributed to the secondary metabolites produced by the endophyte, which could possess biocontrol properties. Several studies have already demonstrated this ability of endophytes to produce antimicrobial compounds that effectively inhibit the growth of bacterial pathogens [62,63] or disrupt bacterial communication mechanisms [63–65]. This is in clear agreement with the compounds tentatively identified by mass spectrometry in the methanolic fungal extract of *A. leptinellae* E138: altechromone A, maculosin, ciclo(L-Phe-L-Pro), and phomopsinone A. The presence of altechromone A, which has been already shown to promote plant growth and inhibit the formation of bacterial biofilms [51,52], could help explain our results regarding the inhibition of bacterial growth and the overall improvement in plant development, regardless of the presence or absence of *P. syringae*

NCPPB 1464. This effect could have been facilitated by the presence of maculosin, known for its antibacterial and antioxidant properties [53], and ciclo(L-Phe-L-Pro), a compound previously found to disrupt QSA [66]. The identification of phomopsinone A, which has demonstrated antifungal potential [55,56], may suggest that the action of *A. leptinellae* E138 is not solely limited to bacterial control but may also be applicable in strategies against fungal pathogens. This finding aligns with previous studies highlighting the role of endophytic fungi as sources of natural bioactive compounds with diverse biological activities [67–71].

In addition to the effect of the antimicrobial metabolites produced by *A. leptinellae* E138, which may directly explain the biocontrol of *P. syringae* NCPPB 1464, other indirect mechanisms may have contributed to this biocontrol and the observed plant growth promotion. All these direct and indirect traits exhibited by *A. leptinellae* E138, such as the production of antimicrobial compounds, phytohormone-like production, phosphate solubilization, and antioxidant activity, could indicate a multifaceted fungal species that could be considered not only as a biocontrol agent but also as a contributor to plant health and disease resistance, a broader concept proposed by Baron and Rigobelo [72]. Improving plant growth, in addition to increasing fruit yield expectations, may also impact plant fitness and consequently its defensive system [73]. Therefore, plant-growth-promoting traits could also be indirectly considered as biocontrol traits. The production of phytohormones, particularly indoleacetic acid (IAA) and gibberellins (GA3), by *A. leptinellae* E138 is consistent with other studies that have highlighted the role of endophytic fungi in promoting plant growth through their synthesis [74], as these phytohormones play a crucial role in regulating plant growth and development. Additionally, the ability of *A. leptinellae* E138 to enhance phosphate solubilization and ammonia synthesis underscores its potential for nutrient mobilization, consistent with the plant-growth-promoting functions of endophytic microorganisms reported in previous studies [75]. Furthermore, the sustained increase in phosphorus solubilization over time may suggest the efficiency of *A. leptinellae* E138 in making phosphorus more available to plants, which is also crucial for plant growth and development [76]. Finally, the total polyphenol content of the *A. leptinellae* E138 extract may indicate its potential to mitigate oxidative stress in plants, which may also contribute to overall plant health and resilience [77].

Based on the results of the mycopriming assay, the *A. leptinellae* E138 extract significantly increased the germination rates of *S. lycopersicum* seeds, regardless of the presence or absence of the pathogen *P. syringae* NCPPB 1464. This suggests that the *A. leptinellae* E138 extract may be an effective germination stimulant in tomato seeds, facilitating more robust seedling establishment, even in the presence of pathogens. This is supported by the significant increases in root length, number of roots, and vigor index of seedlings observed after its application. Seed treatment with beneficial fungi is gaining popularity due to its potential to contribute to more sustainable agriculture. Various articles have already demonstrated its effectiveness in promoting the development of different plant species, both in terms of growth promotion [78] and protection against biotic [79,80] and abiotic [81,82] stresses. In our case, although the results presented demonstrate a lack of phytotoxicity when used on tomato plants, further studies including a wider range of crops and employing different application conditions could allow for evaluating the actual potential of this product for future commercial use. The prospects are promising based on previous findings that have demonstrated the ability of endophytic extracts and microbial inoculants to enhance plant growth and confer resistance to pathogens [83–85].

The postemergence application of *A. leptinellae* E138 extract exhibited significant effects on various aspects of tomato plant health and disease severity under greenhouse conditions, particularly in presence of the pathogen *P. syringae* NCPPB 1464. The significant improvements in shoot and root length, the number of roots, and the vigor index, even in the presence of *P. syringae* NCPPB 1464, underscore the potential of *A. leptinellae* E138 as a growth-promoting endophyte. These findings align with the concept of utilizing endophytes not only for biocontrol but also to enhance overall plant health and fitness.

Additionally, the area under the disease progression curve (AUDPC) values revealed a significant reduction in disease severity in the presence of the endophyte, emphasizing its potential role in mitigating the impact of *P. syringae*-induced diseases on tomato plants. This is consistent with the findings of Mostafa et al. [86], who demonstrated the impairment of virulence and inhibition of quorum sensing of *P. aeruginosa* through the action of different metabolites such as polyphenols from *Salix tetrasperma*. Similarly, Yoo et al. [87] described the potential of the strain *Aspergillus terreus* JF27 for the biocontrol of *P. syringae* in tomato plants, impacting the same parameters observed in our study. Thus, inoculation with the strain significantly reduced disease severity and enhanced shoot length and weight under greenhouse conditions. This outcome highlights the potential of direct application of the fungal extract, which may have a similar effect to that observed with the fungal inoculation. Therefore, the post-emergence application of *A. leptinellae* E138 extract has demonstrated significant potential to improve tomato plant growth and mitigate the effects of *P. syringae*-induced diseases under greenhouse conditions. The results are consistent with previous research on the beneficial effects of endophytic fungi on plant health and disease resistance and underscore the promising role of *A. leptinellae* E138 in sustainable plant disease management strategies.

## 5. Conclusions

Our study revealed the multifaceted potential of *Alternaria leptinellae* E138 as a biocontrol agent against *Pseudomonas syringae* in tomato plants, as well as its role as a plant growth promoter. Filtrates and extracts from this endophyte exhibited remarkable antimicrobial activity, disruption of quorum sensing activity, phytohormone-like production, phosphate solubilization, and antioxidant activity. Several secondary metabolites were tentatively identified in the methanolic extract of the endophyte, including altechromone A, maculosin, ciclo(L-Phe-L-Pro), and phomopsinone A, compounds with either plant-growth-promoting or antimicrobial and antifungal activities. Application of the extract via mycopriming increased seed germination and improved growth parameters of tomato seedlings, regardless of the presence or absence of *P. syringae*. Its postemergence application mitigated disease severity and promoted tomato plant growth under greenhouse conditions. This study provides valuable insights into the field of endophytic research and phytopathology, contributing to sustainable plant disease management in agriculture. Further research should delve deeper into the metabolites involved in biocontrol and incorporate field trials encompassing a wider range of environments, crops, and pathogens.

**Supplementary Materials:** The following supporting information can be downloaded at: https://www.mdpi.com/article/10.3390/horticulturae10040334/s1, Figure S1. ESI-LC-MS mass spectrum extracted from the liquid chromatogram peak (above) and the extracted ion mass spectrum corresponding to the signal of Altechromone A (according to 51,52) at $m/z$ 191.06 (M+H+) (below). Figure S2. ESI-LC-MS mass spectrum extracted from the liquid chromatogram peak (above) and the extracted ion mass spectrum corresponding to the signal of Maculosin (according to 53) at $m/z$ 261.12 (M+H+) (below). Figure S3. ESI-LC-MS mass spectrum extracted from the liquid chromatogram peak (above) and the extracted ion mass spectrum corresponding to the signal of Ciclo(L-Phe-L-Pro; according to 54) at $m/z$ 245.12 (M+H+) (below). Figure S4. ESI-LC-MS mass spectrum extracted from the liquid chromatogram peak (above) and the extracted ion mass spectrum corresponding to the signal of Phomopsinone A (according to 55,56) at $m/z$ 225.11 (M+H+) (below).

**Author Contributions:** Conceptualization, C.G.-L., O.S. and S.R.; methodology, C.G.-L. and S.R.; formal analysis, C.G.-L.; investigation, C.G.-L.; data curation, O.S.; writing—original draft preparation, C.G.-L. and O.S.; writing—review and editing, O.S. and S.R.; supervision, O.S.; project administration, S.R.; funding acquisition, O.S. and C.G.-L. All authors have read and agreed to the published version of the manuscript.

**Funding:** Carlos García-Latorre has been financed by a pre-doctoral grant (PD18037) from the Regional Government of Extremadura (Spain) and by the European Social Fund (ESF).

**Data Availability Statement:** Data are contained within the article and Supplementary Materials.

**Conflicts of Interest:** The authors declare no conflicts of interest.

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
