# Peer review of "Biological Control of Pseudomonas syringae in Tomato Using Filtrates and Extracts Produced by Alternaria leptinellae"

_horticulturae, doi:10.3390/horticulturae10040334_

Round 1

Reviewer 1 Report

Comments and Suggestions for Authors

Title: Biological Control Strategies for Tomato Pathogens: Examining the Role of the Metabolites of Alternaria leptinellae Against Pseudomonas syringae pv. tomato

The study addresses a gap in sustainable agricultural practices by exploring the biocontrol potential of Alternaria leptinellae against Pseudomonas syringae in tomatoes. The focus on endophytic fungi for plant disease management is commendable. The study utilized various assays to examine the direct and indirect mechanisms of biocontrol, including quorum sensing disruption, phytohormone-like substance production, and antioxidant activity.

The title of the study is unnecessarily long.

Although mass spectrometry is used to identify secondary metabolites with antimicrobial properties, the study does not provide a clear link between specific metabolites and their biocontrol mechanisms. A more detailed analysis and discussion on how these metabolites contribute to the observed effects would strengthen the study.

Conduct a more in-depth analysis of the identified metabolites. Exploring the specific biocontrol mechanisms of these compounds could provide valuable insights into their role in disease management.

The study does not clearly specify if appropriate control treatments, such as application of extracts without active metabolites, were used in all experiments. This information is critical for attributing observed effects specifically to A. leptinellae metabolites.

There is insufficient information on the environmental conditions within the greenhouse during the experiment. Factors such as temperature, humidity, and light intensity can significantly influence both plant growth and the activity of biocontrol agents.

Future studies should aim for replication in different geographic locations and under varying environmental conditions to validate the universality and scalability of the findings.

I recommend a comprehensive review and revision of the manuscript's English language to enhance its readability and academic rigor. It would be beneficial to engage the services of a proofreader who is proficient in English, particularly one familiar with the manuscript's academic field.

This study makes a contribution to the field of biological control and sustainable agriculture by highlighting the potential of A. leptinellae metabolites against P. syringae in tomato plants. While the findings are promising, addressing the identified weaknesses and implementing the suggested improvements would significantly enhance the study's validity and applicability. A more detailed examination of control measures, environmental conditions, and the underlying mechanisms of action will provide a stronger foundation for the biocontrol strategies proposed. Further research that builds on this study's findings, incorporating broader environmental conditions and in-depth mechanistic insights, will be crucial for advancing our understanding of sustainable plant disease management.

Comments on the Quality of English Language

I recommend a comprehensive review and revision of the manuscript's English language to enhance its readability and academic rigor. It would be beneficial to engage the services of a proofreader who is proficient in English, particularly one familiar with the manuscript's academic field.

Author Response

The study addresses a gap in sustainable agricultural practices by exploring the biocontrol potential of Alternaria leptinellae against Pseudomonas syringae in tomatoes. The focus on endophytic fungi for plant disease management is commendable. The study utilized various assays to examine the direct and indirect mechanisms of biocontrol, including quorum sensing disruption, phytohormone-like substance production, and antioxidant activity.

The title of the study is unnecessarily long.

Response: shortened

Although mass spectrometry is used to identify secondary metabolites with antimicrobial properties, the study does not provide a clear link between specific metabolites and their biocontrol mechanisms. A more detailed analysis and discussion on how these metabolites contribute to the observed effects would strengthen the study.

Response: the use of mass spectrometry was a first approach for the metabolite identification, but it was preliminary. As indicated in the original version of the manuscript (L477-478), these preliminary results require further research to unequivocally confirm their identification. For this reason, we did not want to go further in the discussion derived of such identification, just in case the results to be obtained in a future not completely agree with those here presented. Even though, a detailed analysis on how those metabolites could contribute to the observed effects has been given on lines 508-525.

Conduct a more in-depth analysis of the identified metabolites. Exploring the specific biocontrol mechanisms of these compounds could provide valuable insights into their role in disease management.

Response: as indicated in the previous comment, the identification of the metabolites in the present study was preliminary and further studies are required to confirm it. Therefore, we did not want to base our discussion in such tentative identification. We thought that the analysis gave in lines 508-525 regarding how those metabolites could have contributed to the biocontrol has been detailed enough at this point of the research. In a future, when the identification can be confirmed, we will design specific experiments with these metabolites to evaluate their exact role in the biocontrol and the analysis in the discussion will be much more deeply performed.

The study does not clearly specify if appropriate control treatments, such as application of extracts without active metabolites, were used in all experiments. This information is critical for attributing observed effects specifically to A. leptinellae metabolites.

Response: The present work did not study the effect of specific A. leptinellae metabolites against the pathogen, but the effect of the filtrate/extract produced by the endophyte under specific in vitro conditions against the pathogen. Therefore, with this objective, our controls were the corresponding solvents without the extract/filtrate. In our present study, the identification of the metabolites was tentative as a first approach to try to explain the mechanisms involved in the observed effects. As indicated before, once the positive effect of the filtrate/extract has been evidenced, the next step in future studies will be the exact identification of the metabolites involved in such effects, and the design of specific experiments with those metabolites to evaluate their exact role in the biocontrol. At that moment, we agree with the reviewer in the fact of including controls of the extracts without active metabolites. We really appreciate the reviewer comment, but to be applied in further studies, when the specific metabolites are evaluated.

There is insufficient information on the environmental conditions within the greenhouse during the experiment. Factors such as temperature, humidity, and light intensity can significantly influence both plant growth and the activity of biocontrol agents.

Response: we agree with the reviewer in the plausible influence of those environmental parameters on plant growth and on the activity of the biocontrol, but we do not completely understand the reviewer comment, as those parameters, temperature and humidity during the experiments, are included in Figure 1. Light intensity was not measured because in the experimental location (Badajoz, South Spain) it is always more than enough for the growth development.

Future studies should aim for replication in different geographic locations and under varying environmental conditions to validate the universality and scalability of the findings.

Response: thank you for the advice. We clearly agree with the reviewer comment. It will be studied further.

I recommend a comprehensive review and revision of the manuscript's English language to enhance its readability and academic rigor. It would be beneficial to engage the services of a proofreader who is proficient in English, particularly one familiar with the manuscript's academic field.

Response: thank you for your recommendation. The English language has been revised again according to this comment.

This study makes a contribution to the field of biological control and sustainable agriculture by highlighting the potential of A. leptinellae metabolites against P. syringae in tomato plants. While the findings are promising, addressing the identified weaknesses and implementing the suggested improvements would significantly enhance the study's validity and applicability. A more detailed examination of control measures, environmental conditions, and the underlying mechanisms of action will provide a stronger foundation for the biocontrol strategies proposed. Further research that builds on this study's findings, incorporating broader environmental conditions and in-depth mechanistic insights, will be crucial for advancing our understanding of sustainable plant disease management.

Response: thank you for your comments.

Reviewer 2 Report

Comments and Suggestions for Authors

Comments to the manuscript “Biological Control Strategies for Tomato Pathogens: Examining the Role of the Metabolites of Alternaria leptinellae Against Pseudomonas syringae pv. tomato” by García-Latorre et al.

General comment

The submitted manuscript analyses the effect of extracellular filtrates/extracts of an endophytic strain of the fungal species Alternaria leptinellae (E138) on the seed germination and plant growth performance of tomato, under greenhouse conditions. The effect of the filtrates/extracts was analyzed in the presence and absence of the bacterial phytopathogen Pseudomonas syringae. In vitro assays were also conducted to evaluate the effect of the filtrates/extracts on the growth of the P. syringae and Bacillus subtilis, on the quorum sensing activity (QSA) of Chromobacterium violaceum, and to evaluate the capability of the E138 strain on phytohormone-like production, antioxidant activity and nutrient mobilization. Finally, five metabolites were identified in the filtrates/extracts which are associated with the antibacterial activity and QSA, seed germination and plant growth promotion.

The authors conclude the filtrates/extracts of the strain E138 contain metabolites which are efficient plant growth promoters and biocontrol agents against Pseudomonas syringae in tomato plants. Such filtrates/extracts can be used as mycopriming agents to promote seed germination and plant growth, or applied postemergence to protect the plant against bacterial pathogens.

The submitted manuscript is well written, the methodology is properly described, the results are clearly explained, and the discussion adequately addressed. Thus, I consider that the document is suitable for its publication in the Horticulturae journal after a minor review. Below are some specific comments for the authors' consideration.

Specific comments:

1. The figure 1 can be considered as a supplementary figure.

2. Please define the acronym GAE in the line 234, before its first use in the line 236.

3. The name of Table 1 can be shortened. I kindly suggest something like “Effect of Alternatia leptinellae (E138) filtrates on the bacterial growth and the quorum sensing activity (QSA).” All experimental details and statistical analysis can be allocated in the table footnote. Figures and table names must be short, auto explicative, and attractive.

4. The control used in QSA experiments ((Z-)-4-bromo-5-(bromomethylene)-2(5H)-furanone) for the results in Table 1 must be specified in the table footnote, as specified for penicillin.

5. I suggest renaming the Table 2 to short and concise name as “Effect of Alternaria leptinellae extract, Pseudomonas syringae, and their interaction on the germination rate of Solanum lycopersicum seeds.” As in the previous suggestion for Table 1, all the experimental details and statistical description can be allocated in a table footnote.

6. The same previous suggestions for tables 1 and 2 about the names and statistical and experimental details apply for Tables 3, 4, and 5.

7. In the text explaining the results of priming experiments you state that the seeds of the Control- group “… exhibited a steady increase in germination rates from day 4 to day 15, reaching a germination rate of around 71% at the end (page 9, lines 369-370).” However, the steady increase in germination appears to reach the maximum value (71%) since the day 9 or 13 (as observed by the superimposition of the standard error bars), with no further increase observed. Please clarify if this appreciation is or not true. If true, I kindly suggest modifying the cited text to be congruent with the results depicted in Figure 2.

8. The same previous comment applies for the results of Control + depicted in Figure 2. But in this case, it appears that the maximum germination rate is reached on day 11. As the previous comment, please clarify and modify the corresponding text if necessary.

9. As in the previous two comments, the maximum seed germination rate appears to be reached on day 13 for E138+ and E138- conditions. Please clarify.

10. The data in Table 5 is difficult to visualize due to the large amount of information it contains. I suggest splitting these data, putting the results of Phosphorus solubilization in an independent Table.

Author Response

General comment

The submitted manuscript is well written, the methodology is properly described, the results are clearly explained, and the discussion adequately addressed. Thus, I consider that the document is suitable for its publication in the Horticulturae journal after a minor review. Below are some specific comments for the authors' consideration.

Response: Thank you very much for your positive comments.

Specific comments:

  1. The figure 1 can be considered as a supplementary figure.

Response: We appreciate this comment, but one of the other reviewers considers the information given in Figure 1 as quite relevant. Therefore, as the manuscript does not present many figures (only two), we think it could be maintained in the main text.

  1. Please define the acronym GAE in the line 234, before its first use in the line 236.

Response: corrected

  1. The name of Table 1 can be shortened. I kindly suggest something like “Effect of Alternatia leptinellae (E138) filtrates on the bacterial growth and the quorum sensing activity (QSA).” All experimental details and statistical analysis can be allocated in the table footnote. Figures and table names must be short, auto explicative, and attractive.

Response: changed

  1. The control used in QSA experiments ((Z-)-4-bromo-5-(bromomethylene)-2(5H)-furanone) for the results in Table 1 must be specified in the table footnote, as specified for penicillin.

Response: specified

  1. I suggest renaming the Table 2 to short and concise name as “Effect of Alternaria leptinellae extract, Pseudomonas syringae, and their interaction on the germination rate of Solanum lycopersicum seeds.” As in the previous suggestion for Table 1, all the experimental details and statistical description can be allocated in a table footnote.

Response: changed

  1. The same previous suggestions for tables 1 and 2 about the names and statistical and experimental details apply for Tables 3, 4, and 5.

Response: changed

  1. In the text explaining the results of priming experiments you state that the seeds of the Control- group “… exhibited a steady increase in germination rates from day 4 to day 15, reaching a germination rate of around 71% at the end (page 9, lines 369-370).” However, the steady increase in germination appears to reach the maximum value (71%) since the day 9 or 13 (as observed by the superimposition of the standard error bars), with no further increase observed. Please clarify if this appreciation is or not true. If true, I kindly suggest modifying the cited text to be congruent with the results depicted in Figure 2.

Response: We agree with this mistake and it has been corrected in the text. Thank you for noticing it.

  1. The same previous comment applies for the results of Control + depicted in Figure 2. But in this case, it appears that the maximum germination rate is reached on day 11. As the previous comment, please clarify and modify the corresponding text if necessary.

Response: clarified. Thank you again.

  1. As in the previous two comments, the maximum seed germination rate appears to be reached on day 13 for E138+ and E138- conditions. Please clarify.

Response: clarified. Thank you again.

  1. The data in Table 5 is difficult to visualize due to the large amount of information it contains. I suggest splitting these data, putting the results of Phosphorus solubilization in an independent Table.

Response: The table has been split into two independent tables, labeled as Table 5 and 6, following the suggestion made by the reviewer.

Reviewer 3 Report

Comments and Suggestions for Authors

The manuscript described using a fungal endophyte, Alternaria leptinellae E138, isolated from the leaves of Ornithopus compressus, to study its biocontrol properties against Pseudomonas syringae pv. tomato. Bacterial growth and quorum-sensing inhibition were investigated. The phytohormone-like substances, nutrient mobilization, and antioxidant activity were assayed to study the plant-growth promotion potentials, agreed with the results from the mycopriming assays and postemergence applications. In addition, four compounds were identified by mass spectrometry, which may contribute to the biocontrol activities found in the A. leptinellae extract. The present work on the study of the biocontrol properties of A. leptinellae is novel and shows that A. leptinellae extract is a potential biocontrol agent to help plant growth and is resistant to pathogen invasion. The manuscript was well-written and organized seriously, including the introduction, materials, and methods. However, some details were missing and needed to be improved before the recommendation. The suggestions are listed below.

Major:

1.     The designation of the Alternaria legionellae E138 was questioned.

-        The information on Genbank accession number KP698337 was inconsistent with the Alternaria legionellae E138 shown in the manuscript. They should be updated to be consistent with current data.

2.     The lack of essential mass spectrometry evidence.

- The mass spectrometry graphs should be provided to show the peaks of the identified compounds.

- How were four compounds, Altechromone A, Maculosin, Ciclo(L-Phe-L-Pro), and Phomopsinone A, identified? Were the standards or known compounds used for comparison?

3. Please explain why Bacillus subtilis was used for the disc diffusion and minimum inhibitory concentration assays shown in Table 1. B. subtilis is not a plant pathogen.

Minor:

1.     The showings of bacterial names should be consistent. For clear expression, Alternaria leptinellae E138, Bacillus subtilis NCTC 8236, Pseudomonas syringae pv. tomato NCPPB 1464 and Chromobacterium violaceum NCTC 9757 could be used. A. leptinellae E138 or E138 could be used as the abbreviation in the text after first mentioned.

2.     Page 5, Line 221. The duration was inconsistent with those shown in the x-axis of Figure 1.

3.     It was not a good way to show the QSA data under the disc diffusion assay column. The table should be modified.

4.     Table 6. Cp04 should be Cpd04 instead.

5.     The positions of the labels that use different letters to indicate significant differences in the tables are usually the superscripts alongside the values.

6.     Tables 5 and 6 were not easy to understand. It is suggested to rearrange the tables.

7.     In the discussion, it is suggested that other potential fungal endophytes with A. leptinellae E138 be compared for the biocontrol of Pseudomonas syringae.

8.     The conclusions need to explain the future perspectives on using A. leptinellae E138 as biocontrol agents on tomatoes and other crops.

Comments on the Quality of English Language

Minor editing of English language required.

Round 2

Reviewer 1 Report

Comments and Suggestions for Authors

The revised version has seen considerable improvements. However, a few minor aspects still require clarification.

Which software was used for statistical analysis?

Conclusion is too long. Concisely conclude your study

Comments on the Quality of English Language

Minor editing of English language required

Author Response

The revised version has seen considerable improvements. However, a few minor aspects still require clarification.

Which software was used for statistical analysis?

Response: All analyses were performed with the Statistix v. 8.10 package (Analytical Software, USA). This aspect has been now incorporated to the text.

Conclusion is too long. Concisely conclude your study

Response: It has been shortened in the corrected versión.

Reviewer 3 Report

Comments and Suggestions for Authors

The manuscript described using a fungal endophyte, Alternaria leptinellae E138, isolated from the leaves of Ornithopus compressus, to study its biocontrol properties against Pseudomonas syringae pv. tomato. Bacterial growth and quorum-sensing inhibition were investigated. The phytohormone-like substances, nutrient mobilization, and antioxidant activity were assayed to study the plant-growth promotion potentials, agreed with the results from the mycopriming assays and postemergence applications. In addition, four compounds were identified by mass spectrometry, which may contribute to the biocontrol activities found in the A. leptinellae extract. The present work on studying the biocontrol properties of A. leptinellae is novel. It shows that A. leptinellae extract is a potential biocontrol agent to help plant growth and is resistant to pathogen invasion. The manuscript was updated and improved according to the suggestions. However, some still needed to be further modified. It is suggested that the manuscript be recommended after further modifications. The suggestions are listed below.

Major:

1.     The designation of Alternaria leptinellae E138 was questioned because the information on Genbank accession number KP698337 was inconsistent. It should be updated to be consistent with current data. The author’s reasons were acceptable. The information on Genbank KP698337 is suggested to be updated to show consistency.

2.     The mass spectrometry graphs were provided to show the peaks of the identified compounds. However, it needs the following improvements.

- The title of the supplementary material should indicate the strain number as follows: Mass spectrometry graphs associated with the compounds tentatively identified in the extract of Alternaria leptinellae E138.

- The supplementary material should be part of the article and described in the manuscript's back matter, the Supplementary Materials.

3. The authors explained that four compounds, Altechromone A, Maculosin, Ciclo(L-Phe-L-Pro), and Phomopsinone A, were identified preliminarily. The reference could also be cited in the legends of the supplementary materials.

4. The reasons why Bacillus subtilis was used for the disc diffusion and minimum inhibitory concentration assays could be explained in Section 3.1 because B. subtilis is not a plant pathogen.

Minor:

1.     Section 2.1 The scientific uses of bacterial names and isolates should be consistent. For example, Bacillus subtilis NCTC 8236, Pseudomonas syringae pv. tomato NCPPB 1464 and Chromobacterium violaceum NCTC 9757 could be used. The brackets did not need to indicate the bacterial isolate numbers after the scientific names.

2.     Section 2.6. The reasons why the duration was inconsistent with those shown in the x-axis of Figure 1 could be described in Section 2.6.

Author Response

The manuscript was updated and improved according to the suggestions. However, some still needed to be further modified. It is suggested that the manuscript be recommended after further modifications. The suggestions are listed below.

Major:

  1. The designation of Alternaria leptinellae E138 was questioned because the information on Genbank accession number KP698337 was inconsistent. It should be updated to be consistent with current data. The author’s reasons were acceptable. The information on Genbank KP698337 is suggested to be updated to show consistency.

Response: We will have this comment into account and we will try to update the information of our isolate KP698337 on GenBank as soon as possible.

  1. The mass spectrometry graphs were provided to show the peaks of the identified compounds. However, it needs the following improvements.

- The title of the supplementary material should indicate the strain number as follows: Mass spectrometry graphs associated with the compounds tentatively identified in the extract of Alternaria leptinellae E138.

Response: Indicated

- The supplementary material should be part of the article and described in the manuscript's back matter, the Supplementary Materials.

Response: Done

  1. The authors explained that four compounds, Altechromone A, Maculosin, Ciclo(L-Phe-L-Pro), and Phomopsinone A, were identified preliminarily. The reference could also be cited in the legends of the supplementary materials.

Response: Indicated

  1. The reasons why Bacillus subtilis was used for the disc diffusion and minimum inhibitory concentration assays could be explained in Section 3.1 because B. subtilis is not a plant pathogen.

Response: Explained

Minor:

  1. Section 2.1 The scientific uses of bacterial names and isolates should be consistent. For example, Bacillus subtilis NCTC 8236, Pseudomonas syringae pv. tomato NCPPB 1464 and Chromobacterium violaceum NCTC 9757 could be used. The brackets did not need to indicate the bacterial isolate numbers after the scientific names.

Response: Corrected

  1. Section 2.6. The reasons why the duration was inconsistent with those shown in the x-axis of Figure 1 could be described in Section 2.6.

Response: described